# Differences of Modality Use between Telepractice and Face-to-Face Administration of the Scenario-Test in Persons with Dementia-Related Speech Disorder

**DOI:** 10.3390/brainsci13020204

**Published:** 2023-01-26

**Authors:** Mirjam Gauch, Sabine Corsten, Katharina Geschke, Isabel Heinrich, Juliane Leinweber, Bianca Spelter

**Affiliations:** 1Department of Psychiatry and Psychotherapy, University Hospital Mainz, 55131 Mainz, Germany; 2Faculty of Healthcare and Nursing, Catholic University of Applied Sciences Mainz, 55122 Mainz, Germany; 3Faculty of Engineering and Health, University of Applied Sciences and Arts Hildesheim/Holzminden/Göttingen, Health Campus Göttingen, 37075 Göttingen, Germany

**Keywords:** telepractice, primary progressive aphasia, modality use

## Abstract

Telepractice is increasingly finding its way into the care of people with dementia. Web-based delivery of speech and language therapy (SLT) is feasible and has the potential to improve communication in people with dementia-related speech disorders. Although experts are discussing the strengths and weaknesses of telepractice, a precise analysis of the differences between analogue and digital communication for this heterogeneous group of patients is still missing. The three current single cases investigated verbal and nonverbal aspects of communication in a face-to-face (F2F) and digital setting through a qualitative research design. Using the scenario-test (ST) in person and via big blue button (BBB; video conferencing system), several decisive factors were detected, influencing the effectiveness of communication in a F2F compared to a digital setting. The most important results of the qualitative content analysis are described for each case individually. Additionally, the influence of person-related factors, such as age, diagnosis, presence of depression, and level of education, is presented. Perceptual, executive, and affective disorders, as well as aids of relatives, are considered separately. The results indicate that executive functions, affects, and perceptual deficits need to be taken into account if telepractice is to be applied. Age, education, and distinct forms of dementia might be decisive for successful telepractice as well.

## 1. Introduction

Speech disorders occur particularly in degenerative, cortical diseases, for example, primary progressive aphasia (PPA) or Alzheimer’s disease (AD). While the language disorder is secondary to memory deficits in AD, it is the primary and central symptom of PPA [1]. According to Alzheimer’s Disease International (2021), it is estimated that 55 million people worldwide are affected by dementia and, due to demographic change, its incidence is expected to increase in future. Although PPA is known as a rare condition, the incidence of 3:100.000 [2,3] shows an upward trend, likely due to increasing awareness of the diagnosticians [4]. Symptoms of PPA begin insidiously and develop a severe and progressively global profile in progress [5]. PPA can be classified into the non-fluent (nfvPPA), semantic (svPPA), and logopenic (lvPPA) variants [6]. The svPPA variant is most frequent, contributing to approximately 39% of all diagnoses [7]. Those affected show deficits in naming and single-word comprehension. The lvPPA is the second most common variant, with about 30% representation [7]. Patients with these variants show word-finding disorders in spontaneous speech and naming, as well as impairments in repeating sentences and phrases. nfvPPA accounts for 18% of patients with PPA [7]. Linguistically, it is characterised, in particular, by agrammatism in expressive speech and/or speech strain with inconsistent phonetic errors and misalignments [6,7]. According to clinical criteria, about 50–70% of dementia patients are classified as having AD [8]. The linguistic profile of AD consists of word-finding disorders, stereotypies, and simplified syntax. In the course of the disease, echolalia and confabulations, as well as a loss of discourse coherence, are common [1]. The progressive speech disorder leads patients with PPA and AD into a dependency on aided communication or empathetic family carers [9,10]. For this reason, the use of different modalities, e.g., writing, gesturing, drawing, or usage of communication aids, is of essential importance [11].

The use of written language may be impaired, particularly in the context of surface dyslexia/dysgraphia in svPPA [6]. However, errors in reading and writing also occur in nfvPPA, lvPPA, and AD [1,12]. At the same time, resources in this area can offer good compensation, especially if performance in written naming is superior to oral naming [13].

Drawing may be limited in svPPA patients with advanced semantic deficits or in AD because of visual–spatial deficits [8]. However, a single-case study by Murray (1998) shows how a patient with nfvPPA benefited from the ‘back to the drawing board’ approach [14]. During therapy, she improved her drawing skills and acquired a form of communication that she spontaneously used as a form of expression when her verbal skills continued to deteriorate. Murray (1998) suggests that this approach was pleasant for the patient due to its strength-based focus.

The use of gestures may be limited by motor skills deficits that occur in the context of apraxia or concomitant parkinsonism [12,15]. Nelissen and colleagues (2010) describe a sample of patients with PPA who showed marked difficulties with the interpretation and imitation of gestures [16]. Deficits in the use of gestures were thought to be caused by volume reductions in the perisylvian parietotemporal region. According to Fried-Oken et al. (2011), however, even patients in mid-stage of PPA can benefit from the use of gestures [11]. For patients in late-stage AD, communicating non-verbally via facial expressions, gestures, or eye contact is recommended [17,18].

The use of communication aids may be impaired in the context of PPA when executive dysfunction or impairments in other cognitive domains are present. However, especially in svPPA, patients benefit from their intact procedural memory, which facilitates, for example, the use of a smartphone and allows compensation via high-tech strategies [19]. A communication book can help to compensate communicative breakdowns and supplement speech if needed [20]. Experts emphasise the importance of individually adapted communication books that cover personally relevant topics [20].

To assess communicative–pragmatic skills, appropriate measurement instruments have to be applied [21,22]. Although no procedure for testing communicative skills has reached the consensus criterion of >70% in the Research Outcome Measurement in Aphasia (ROMA) consensus statement yet, the scenario test (ST) was most frequently chosen (by 57% of participants) to determine this construct [23]. The ST can be used to directly measure language use in everyday life [24]. While the examiner and the examinee enter into dialogue, it is assessed how much support the examinee needs to cope with a communication situation [25]. The test was developed for people with severe post-stroke aphasia. A transferability to dementia is assumed by the authors of the ST [25]. As the communicative behavior of patients with PPA and stroke-related aphasias is similar [26], the ST could be a valid measurement tool for PPA diagnostics as well.

Due to the ongoing SARS-CoV-2 pandemic, health services are increasingly using telepractice to maintain patient care [27,28]. The use of telepractice in people with neurodegenerative diseases is of particular importance, as they have a high vulnerability due to their age and must be given special consideration when implementing infection control measures [29]. The fact that PPA is still a rare syndrome makes telepractice even more significant, as patients do not have to travel long distances or incur travel costs to find an expert in this field [30,31]. A pilot study demonstrated the feasibility of telepractice in people with PPA and other dementia diagnosis and showed significant improvements in communication confidence after eight weekly interventions [32]. The authors highlight delays or reduced sound quality as some of the limitations of their web application, as they affected verbal communication. Studies indicate that patients in telemedicine settings tend to engage in a lower proportion of conversation, while competent medical professionals engage in a higher proportion of verbal conversation compared to a face-to-face (F2F) conversation [33]. In a study on online support groups for patients with dementia and their relatives, group members reported that they found it difficult to follow the turn-taking because the videos of the people speaking were sometimes small and difficult to see [34]. In addition, it was reported to be more difficult for the patients to identify emotions in the video conference [34]. It could be assumed that if the video quality is particularly poor, people use fewer gestures, as participants in a 1995 Australian study in a video conference used fewer gestures because fast movements contributed to poorer image quality [35]. In addition, pointing at objects and using gestures might be less successful in the digital setting because objects are out of focus or outside the camera image [36]. Furthermore, studies indicate that poor audio and video quality in telediagnostics impairs understanding between the test person and the examiner [37,38] Even though technical possibilities have improved in the last years, recent and previously unpublished research indicates differences in modality use in a digital or F2F environment.

For this, it can be hypothesized that the setting influences communication because perception may be impaired in telediagnostics [32]. Regardless of the setting, the presence of hearing impairment might affect verbal performance [39], and perceptual deficits in general can influence the selection of modalities [40]. Executive disorders can be hypothesized to impair communicative performance, as individuals with aphasia and impaired cognitive flexibility used a lower variation of communicative expressions in previous research [41]. Furthermore, it can be hypothesized that an underlying pathology of dementia syndrome, presence of depression, and age have an impact on affects. There is evidence in the literature that patients affected by svPPA, comorbid depression, young age of diagnosis, and early stage of the disease show negative affects and high levels of distress in particular [42]. The age might influence communicative abilities, as well, because of the physiological brain decline [43] and correlated changes in language processing and naming performance [39,44,45]. As with age, a person’s level of education influences their communicative abilities [46]. Finally, it can be hypothesized that supportive communication partners have a positive impact on communication. Interlocutors who interact with the person with aphasia without causing them to lose face have been identified in studies as particularly supportive. Other helpful behaviors include clarifying sequences, acknowledgement, and reassurance [47].

This study examines differences in the modality use of participants with dementia-related language disorders using the ST in a F2F and telediagnostic setting. The qualitative analysis was intended to work out the special features of the sample of people with dementia-related language disorders in dealing with the test material of the ST in telediagnostic and F2F setting. In addition, initial reflections will be made on the causes of communication disorders in analogue and digital settings.

## 2. Materials and Methods

The current diagnostic study included three single-cases and used a qualitative research design. The three single cases were extracted from a dataset of 21 people due to inclusion criteria, which are described below. The diagnostic was conducted in the Memory Clinic of the Department of Psychiatry and Psychotherapy (University Medical Center of the Johannes Gutenberg University Mainz) in the period from January to March 2022. Approval by the Ethics Committee of the Federal Medical Association of Rhineland-Palatinate had been granted in November 2021. The testing was carried out by a speech and language therapist (SLT) using the ST. Relatives or outpatient SLTs were asked to help with setting up the technique if needed and to remain within earshot during the examination to mediate between the subject and the examiner in case of technical difficulties (e.g., disconnection, errors in transmitting visual and auditory information). Quantitative data (e.g., point values) of the diagnostics were compared in order to assess equivalence of digital and analogue testing. The results of the quantitative analysis will be published separately. Qualitative data (e.g., modality use) were assessed through video recording. The results of qualitative content analysis [48] in three outstanding participants will be described below.

### 2.1. Assessment

The ST measures multimodal communication in an interactive setting between test person and SLT. The test items are presented in black and white pictures. Subsequently, the patients were asked to put themselves in the role of the main person depicted in the image and to communicate as it would be appropriate in the situation [49]. The focus of the diagnostics is the transactional function of communication. Therefore, an item is evaluated according to whether the relevant information was communicated [25]. The ST contains one practice scenario and six test scenarios. Each scenario consists of three pictures of an everyday situation that is intended to elicit communication. The task is given verbatim by the test manual and presented to the test person. During the verbal instruction, the black and white pictures are presented, which promote comprehension in the communication situation. The page is then turned to avoid the respondent pointing to the diagnostic document while formulating their answers. If the content is not transmitted successfully, help is offered. First, the general question is asked: “Can you make it clear in another way?”. If this request does not provide sufficient help, the individual communication modalities are stimulated one after the other: “Can you perhaps make a gesture?”, “Can you perhaps draw it?”, “Can you write it?”, “Is it in your communication book?”. If the transmission of information is still not successful after these aids, the examiner asks yes-no questions that are listed for the respective item on the record sheet (e.g., for item 6c: “Is the soup too hot?”; “Would you like a spoon?”). The video recording of the test situation enables a subsequent evaluation with a simultaneous collection of quantitative and qualitative results. At the item level, it was calculated how much help was needed to complete each item. The highest possible total value was 54. At the proposition level, verbal and non-verbal responses were coded in terms of their modalities. Spontaneously correct responses were coded 2, responses that were successful when prompted were coded 1, and unsuccessful responses were coded 0. Spontaneous reactions that were not successful were coded SNE (spontaneously unsuccessful) [25].

### 2.2. Procedure

The test procedure corresponded to the manual instructions of the ST in both the F2F and analogue settings [25]. The time allowed for the test was 15–45 min. The video camera was set up in a corner of the room so that the examiner and the subjects’ facial expressions and gestures could be recorded at the same time during the diagnostic process (Figure 1). During F2F and telediagnostics, the use of a personal communication book was allowed.

The F2F diagnostic took place in an examination room in the Memory Clinic. The room had with limited distractions and was in a quiet environment so as not to interfere with the participants’ attention. The test person took a seat at the table opposite the examiner. Paper and pen were provided. The ST test booklet was also placed on the table.

The telediagnostic took place via the video conferencing system Big Blue Button^®^ (BBB, invokable GmbH, 2022) to ensure data safety and, therefore, compliance the General Data Protection Regulation (DSGVO). In preparation for the telediagnostic, the test persons were sent an individualised invitation via URL to the BBB^®^ study room by email. They could open this at home from their private computers or from another end device (from a speech therapy practice/Memory Clinic) and access it without prior server installation. Video conferencing via BBB^®^ made it possible to share uploaded slides while interlocutors could still see each other via video. The presentation contained the test items of the ST, as well as a white slide between each item to avoid subjects pointing at the pictures, according to the ST’s hand instructions. Participants, relatives, and supporting SLTs were asked to have paper and a pen ready in order to enable multimodal communication in telediagnostics. If a communication book was available, it was also provided.

### 2.3. Qualitative Data Extraction

The three single-cases were analysed using a quantitative analysis procedure [48]. The process of qualitative content analysis (see Figure 2) was carried out in accordance with Kuckartz (2018) [48]. 

The qualitative analysis began with a planning phase in which video material of all recruited participants were watched by one SLT. On the basis of these observations and the existing theories, it was hypothesised that executive dysfunction, perceptual deficits, and negative affect may have a negative influence on verbal and non-verbal communication. In a second hypothesis, it was assumed that person-related and environmental factors affect verbal and non-verbal communication. Included criteria for selected cases were defined. In order to answer the research question, videos of participants who showed a difference in modality use between telediagnostics and F2F diagnostics were to be evaluated using video interaction analysis [45]. These selected cases should show different types of dementia syndromes in order to take into account the heterogeneity of the participants. In a developmental phase, examination videos were transcribed by the SLT, who is, as well, the first author of this paper, according to the rules of the extended content-semantic transcription of Dresing and Pehl (2018) [50]. During transcription, the SLT made notes about behavior and interaction of the participants to capture non-verbal communication. An initial coding system was developed, and definitions of codes were formulated. At this time, the following codes were defined based on the current literature: (1) verbal, (2) non-verbal, (3) perceptual, (4) aid use, (5) executive functions, (6) memory, (7) affective, (8) influence of relatives, and (9) technical problems. Definitions of the initial, inductive codes were developed and discussed within the research team. The research team remained open regarding necessary changes in the code system according to a deductive approach in the course of the qualitative research study. The final coding system can be seen in the results.

### 2.4. Qualitative Data Analysis

In order to make the evaluation more comprehensive, an independent person was then integrated into the evaluation coding process. The coder was not completely blinded. In order to carry out the coding, she had been informed about the intentions of the study. However, the fact that the coder did not know the subjects allowed for a more objective view of behaviours and a more sensitive interpretation of the linguistic utterances. This person was trained as a coder in a test phase, and the category system was tested on parts of the material. If necessary, codes were changed at this phase if there was consensus between SLT and the coder. The material was fully coded by one coder in the coding phase using MAXQDA^®^ a software for qualitative data analysis (VERBI Software, 1989–2021 [51]). The SLT again matched the coded material with the videos in order to directly detect and eliminate possible misinterpretations of the non-verbal communication. Finally, in an evaluation phase, the data matrix was qualitatively analysed. The results of the analysis are described below.

## 3. Results

### 3.1. Participants

Of 21 participants, 29% (n = 6) dropped out before the first examination for personal reasons. Of the 15 remaining participants, three cases were selected according to the criteria described above. Taking these criteria into account, the selection fell unintentionally on three men. All three participants went through the F2F diagnostic first and then the telediagnostic. Since the evaluation of the quantitative results did not show any learning effects, it is assumed that the order had no influence. An overview of the three single-cases and their person-related data are given below (see Table 1). 

### 3.2. Category System

The nine codes formulated in advance were adapted according to a deductive approach in the course of the qualitative research study (see Table 2). 

A new code was established for metalinguistic comments as an expression of excessive demands. Statements, such as “I’m unable to think clearly” were considered noteworthy, but could not be classified as affective, since no conclusions can be drawn from pure observation as to whether it was an emotional expression or a neutral observation by the participant. In addition, the code “memory” had been renamed “working memory”, since only working memory is addressed in the context of the test performance of the ST. According to the model of Baddeley and Hitch (1992) [52], working memory is a constituent of executive functioning. Furthermore, due to the frequent errors in this area, a special code was created called “Difficulties in Role Taking”. This was also assigned to the code “executive”, as the examiner suspected deficits in cognitive flexibility and at the level of the theory of mind as the background to the difficulties. The most important results of the qualitative content analysis are described for each case individually. In the listed significant quotes, the examiner is abbreviated with the letter “E”. The orginal quotes in german language can be seen in Appendix A. The area of the assigned codes is visualised by the document portraits. The frequencies of the different codes are listed as document variables.

### 3.3. Participant 1 (P1)

P1 is a 66-year-old man and a skilled mechanic who lives with his wife in their own home. With the help of occupational therapy and physiotherapy, P1 is always encouraged to engage in activities. According to the medical report, P1 had significant hearing loss, but depression was not present. Anamnestically, the speech-related symptoms had already been observed for five years.

The history, clinical symptoms, course of the disease, as well as the findings of the neuropsychological examination, revealed deficits in several cognitive domains (mental flexibility, semantic and phonetic word fluency, verbal memory, orientation, speech production, and comprehension, as well as delayed recall of figural-constructive memory contents). During the speech examinations, severe word-finding disorders, a slight agrammatism with a strained speech flow, as well as a moderate disorder of speech comprehension, could be objectified. In the absence of additional diagnostic evidence (laboratory, cMRI) for competing causes, a non-fluent variant of primary progressive aphasia (PPA) was diagnosed according to the current diagnostic criteria [6], taking into account the FDG-PET findings with left temporo-parietal and mild frontal glucose hypometabolism and the liquor diagnostic findings. The underlying pathology was found to be AD.

Between 2019 and 2020, P1 participated in psychotherapeutic group interventions at the Memory Clinic of the Department of Psychiatry and Psychotherapy (University Medical Center of the Johannes Gutenberg University Mainz). During this time, the participant’s language performance was mostly stable. Especially in treatment-free periods (e.g., during the lockdown in spring 2020), a worsening of symptoms took place. The participant’s wife took part in the local PPA support group for caregivers and took advantage of the offer of personal counselling following the ST examinations.

Communication in both diagnostic settings was only possible with the support of the communication partner. Yes-no-questions were an effective aid, even if the answers were not always appropriate. In F2F diagnostics, the participant succeeded in conveying 8.3% of the propositions. In addition to speaking, drawing was successfully used as a modality. In telediagnostics, only 4.2% of the propositions were transmitted. Non-verbal communication was not used to compensate. This difference in modality use is examined qualitatively in the following to gain insight into influencing factors.

#### Qualitative Analysis in the Case of P1

The overview of the used codes in the case of P1 (see Figure 3) showes the sequence of codings weighted by the length of the segments. The different codes are presented in different colours. 

In the case of P1, aphasic symptoms were central in both settings. According to qualitative analysis, code verbal (purple) occurred slightly more frequently in telediagnostics (97 vs. 92 codes). P1 showed numerous stereotypies, speech automatisms, and recurring utterances (such as “do-do”, “flawless”, “today”, “seen from that point of view.”). While answering the yes-no questions, indications of deficits in language comprehension emerged.


*E: “I will ask you a few more questions. You may answer yes or no. When answering, think about the picture you have just seen and answer the questions in the same way as the situation is depicted. Do you have a headache?”*

*P1: “Yes… perfectly. Yes”*
(Telediagnostics at P1: 175–176) 

Executive deficits (green) were observed less frequently in telediagnostics compared to F2F diagnostics (5 vs. 2 codes). Sometimes a classification in the domain of executive function, as well as in the domain of aphasic symptoms, was possible:


*E: “For example, you could make a gesture (…)”*

*P1: “Yes?”*

*E: “Stop Stop!” (raises arm as well as right hand, extends palm towards the screen.)*

*P1: (looks towards the hand) “Five. Yes, exactly”*
(Telediagnostics at P1: 45–48)

Perceptual deficits (orange) occurred slightly more frequently in F2F diagnostics compared to telediagnostics (5 vs. codes). Phonematic paraphasias were sometimes difficult to distinguish from perceptual deficits due to the close link between phonological input and output.


*E: “Hm. Ok. I’ll ask another question. Are you looking for a [hεmt] (shirt)?”*

*P1: “A [hʊnt] (dog)?”*
(F2F at P1: 69–70)

The code nonverbal (red) occurred more frequently in F2F setting (19 vs. 3 codes). In the F2F examination, the participant resorted more frequently to the modality “drawing”, although this was used only once “spontaneously successfully”.


*(Dimly draws people with oversized jumpers, similarity to item template clearly recognisable)*
(F2F at P1: 76–76)

In both settings, the participant shows affects (light blue) in the form of spontaneous laughter or sighing. Affects occurred slightly more frequently during F2F diagnostics (20 vs. 18 codes). Mostly, affective codes were assigned in relation to failed verbal and non-verbal communication.


*(E. puts the sheet in front of A6)*

*P1: “Do-o-o-o!” (draws, sighs) “I don’t know.”*
(F2F at P1, pos. 164)

The code aids of relatives (deep blue) were not used in the F2F setting, but they were prominent at the beginning of the telediagnostics setting (see Figure 3). The participant’s wife entered the room several times during telediagnostics to help her husband, who felt overwhelmed using the technology. The fact that the wife spoke reassuringly and with a simplified syntax to her husband made it possible to continue the telediagnostics.


*P1: “And do, today?” (unintelligible)*

*P1′s wife: “That’s all right, darling. Just leave it and listen to what Mrs Gauch tells you. Don’t do anything about it. Don’t pretend anything. Okay? Good.”*

*P1: (unintelligible, smiles, leans forward towards the screen, wife disappears from view).*
(Telediagnostics at P1, pos. 20–22)

### 3.4. Participant 2 (P2)

P2 is a 79-year-old man and a skilled mechanic who worked as a car salesman and lastly worked as a head of a marketing department responsible for employees until early retirement at the age of 60. P2 lives with his wife in their own home. They have two children, but no outpatient help. According to the medical report, a depression is not present. Anamnestically, the speech related symptoms had already been noticed for three years.

The history, clinic symptoms, course of the disease, and neuropsychology with deficits in several cognitive domains (alternation, verbal as well as figural memory, processing speed, word fluency, spatial ability) prove a mild dementia syndrome, which must be assigned to an Alzheimer’s disease with late onset due to the cognitive deficit profile with leading amnestic deficits and the structural cerebral imaging. There are no indications for other causes on imaging and laboratory chemistry. During speech diagnostics, the participant’s symptoms could be classified into the profile of a lvPPA. Although the participant described his speech disorder as clinically leading, the speech disorder was not objectifiable the first and central symptom. Additionally, profound cognitive deficits were present in the initial phase. According to the diagnostic criteria of the current S3 guideline, a probable Alzheimer’s disease was diagnosed.

In the F2F diagnostics, the participant succeeded in transmitting 79.2% of the propositions. The contents were understood by the examiner solely based on the modality of speaking. Occasionally, gestures and drawings were used to accompany the speech. In telediagnostics, 95.8% of the propositions were transmitted. In this setting, too, the information was conveyed exclusively verbally. The use of gestures to accompany speech was reduced in telediagnostics compared to F2F diagnostics.

#### Qualitative Analysis in the Case of P2

In the case of P2, verbal (purple) was the most frequent code (see Table 3), but not the most serious in terms of area (see Figure 4). Compared to P1, the amount of verbal codes was much lower. P2 showed numerous embolophonias/embolophrasias mostly in connection with word-finding disturbances, e.g., “this uh uh Taxi” (telediagnostics at P2: 24). A direct comparison between telediagnostics and F2F diagnostics showed that embolophonias/-phrasias occurred more frequently in telediagnostics than in F2F diagnostics (33 vs. 24 codes). Further semantic paraphrases were observed.


*P2: “Um, yes, tell me: Doesn’t the certificate [actually: prescription] say that I want something liquid? Not necessarily for intoxication, but to get over the cough better.”*
(Telediagnostics at P2: 58)

In the document portrait (see Figure 4), executive deficits (green) take up the largest share of the transcript space, which shows the qualitative importance of symptomatology for the present case. The frequency of codes was higher in the F2F than in the telediagnostic setting (13 vs. 9). The executive deficits manifested themselves in excessive speech production. Nevertheless, the contents of the scenarios could be successfully conveyed.


*P2: “Yes, I don’t know, did we also order the Lus soup? We actually just wanted to eat (laughs) Schni-Schnitzel. But it doesn’t matter. The cutlery isn’t perfect at all, so maybe you can go back again. And then bring us here? (brings hands together gesticulating at chest level, forms a rhombus of fingers, then lowers hands again) Well, I’m not actually and probably (points to the right) you’ve now hired me as a revolutionary and an [other]. I’m actually always polite, as long as no one gives me any guff (gestures unspecifically) (drops his hands) so I’ve still managed to keep that. Then you can also manage it with just a few words. That’s my opinion (points to himself). You certainly don’t see it differently.”*
(F2F with P2: 182)

The code nonverbal occurred more frequently in F2F diagnostics compared to telediagnostics (15 vs. 6 codes). However, in both settings, information was transmitted through verbal communication so that gestures were only used in a supportive and not compensatory way. The document portrait shows a similar area of nonverbal (red) codes in both settings. This is due to two sequences at the end of P2’s telediagnostic transcript in which the examiner explicitly stimulated the use of modalities.


*E: “Would you make that clear in another way?”*

*P2: “Yes…” (stands up in front of the PC) “I would stand up.” (extends right arm) “Give the hand” (shakes right hand and raises left arm so that hand is at shoulder level) “put the arm on the shoulder and say: And next time you come to me.”*
(Telediagnostics at P2, pos. 81–82)

Disturbances in perception (orange) only occurred in telediagnostics (4 vs. 0 codes). All coded segments indicated auditory difficulties; evidence of visual limitations was not coded.


*E: (…) “You have been coughing (makes gesture and sound to “cough”) for weeks. You (points with the cursor to the person in the scenario) go to the doctor. What’s wrong?, he asks.”*

*P2: “What, what, what does he ask? Sorry. I didn’t understand them. [Double lot?]”*
(Telediagnostics at P2, pos. 31–32)

### 3.5. Participant 3 (P3)

P3 is a 68-year-old man and skilled chef who worked as a department manager for various companies. He worked full-time until retirement at the age of 58. P3 lives together with his wife. He has a daughter and a grandchild. At the beginning of the study, the participant received weekly speech therapy and intended to receive occupational therapy in the future. According to the medical report, the participant has been suffering from recurrent depressive episodes for over 10 years. The last psychotherapeutic treatment was five years ago. Anamnestically, the speech-related symptoms had already been noticed for two years.

According to the participant, the initial symptom was confusion between proper names and objects (consistent with anomia). Clinically, however, the leading symptom was memory impairment. The reported deficits could be objectified by test diagnostics. In the course of a neuropsychological examination, significant deficits in verbal, figural episodic memory and phonematic word fluency, spatial ability, verbal working memory, tonic alertness, and divided attention could be detected. Thus, the criteria of multiple-domain amnestic mild cognitive impairment (MCI) are fulfilled [8]. The speech diagnostics revealed the typical findings of a semantic variant of PPA. The main criteria of impaired naming and impaired single-word comprehension were fulfilled. Of the secondary criteria, the required three were fulfilled with impaired object knowledge, intact repeat speech, and intact speech production. In the synopsis of the results, the presence of a svPPA is probable. In the course, a new REM sleep disorder, extrapyramidal symptoms in the form of right-sided rigidity, bilateral bradykinesia, and postural tremor, as well as postural instability in the sense of a hypokinetic-rigid syndrome, developed. The underlying pathology was found to be frontotemporal lobar degeneration.

In F2F diagnostics, the participant was able to convey 91.7% of the propositions. Verbal language was used in 83.3% of the tasks, writing in 4.2%, and a combination of speaking and gestures in another 4.2%. In telediagnostics, 79.2% of the propositions were transmitted. In this setting, speaking was the only modality used.

#### Qualitative Analysis in the Case of P3

P3 showed superficial executive deficits (green) in the telediagnostic and F2F setting. This can be seen in the length of the codes in relation to the total document (see Figure 5), as well as the total amount of codes per setting (F2F diagnostics: 41, telediagnostics: 38). In some test scenarios, the assumption of the role is not successful, which can be attributed to limitations in the understanding of instructions.


*E: (…, turns to item 1a) “You may look at the picture first (short pause, points to person in item 1a) This is you again. You are in a clothes shop. You want to buy a new jumper. The shop assistant approaches you and asks: How can I help you?”*
*(turns to white intermediate page)*

*P3: “How can I help you? What colour should the jumper be and what thickness, so that it is nice and warm”.*
(F2F at P3, items 25–27)

During the assessments with the ST, P3 showed some expressions of being overwhelmed and being nervous, and P3 sometimes verbalised deficits in attention and concentration (pink). These expressions occurred more frequently in the F2F interview (13 codes vs. 1 code).


*P3: “I’m a bit too nervous for the whole story now”.*
(F2F at P3: 198–198)

Negative affect (light blue) was found in connection with the expression of excessive demand, especially in F2F diagnostics.


*P3: (quietly) “I can’t concentrate (raises left hand to forehead, adjusts glasses, looks at test folder, sighs) no, I can’t cope… “I’m unable to think clearly” [had already had] like we tested over there.”*
(F2F at P3: 104)

P3 used aids (yellow) during the remote diagnosis with paper and pencil to compensate for deficits in working memory.


*P3: (has eyes on table in front of him, takes notes while the scenario is shown/creates memory aid).*
(Telediagnostics at P3: 54–54)

The code verbal (purple) occurred more frequently in telediagnostics compared to F2F diagnostics (33 vs. 23 codes). P3 showed semantic paraphasias and iterations. Further, some sequences were unintelligible.


*P3: “Mrs Müller, have you ever handled [actually: ironed] silk blouse clothing (unintelligible)? Can you do that? Then I would entrust them to you then.”*
(Telediagnostics at P3, item 110)

The code perceptual (orange) was used once in the F2F diagnostic. Coding as an executive disorder was also considered for that sequence but discarded due to the immediate request for repetition.


*E: “Then you have found the right jumper. Now you would like to know how much the jumper costs (gesture for “money”).”*



*P3: “Ok. (nods his head, points the biros at the test pad).”*



*E: “But you can’t find the price tag. What do you do?” (turns to white intermediate page)*



*P3: (indistinct, reaching out towards the test folder)*



*E: “Yes. I’ll repeat it all over again. …”.*
(F2F at P3, items 79–83)

### 3.6. Comparison of the Qualitative Findings

The following summary table shows the relative frequencies from F2F and telediagnostics for each subject (see Table 3). It can be seen that aphasic symptoms were coded most frequently in two of the three subjects (P1, P2). In the case of P3, executive deficits were predominant. The transcripts of P1 contained the most non-verbal abnormalities. In addition, he was the only participant who had to obtain help from his wife during the telediagnostics. P2 showed the most affective reactions in the F2F setting. He used less nonverbal communication and showed less executive deficits in telediagnostics compared to F2F diagnostics. In the case of P3, the most expressions of excessive demand were observed. P3 was the only participant who used an aid in order to compensate memory deficits.

## 4. Discussion

The aim of this study was to explore differences in modality use of patients with dementia-related speech disorders within the framework of an overarching project comparing telediagnostic and F2F administration of the ST in this population. Nine codes had been derived from existing evidence. In the process, those were adapted to the following eight: (1) verbal, (2) expression of excessive demand, (3) non-verbal, (4) perceptual, (5) aid use, (6) executive functions, (7) affective, and (8) aids of relatives. Person-related factors, such as age and level of education, as well as environmental factors, such as the role of relatives, were taken into account within the framework of a qualitative evaluation. The influence of a telediagnosis or a F2F interview was particularly interesting, as a direct and intrapersonal comparison could be drawn due to the cross-over design.

A relation between executive deficits and difficulties in verbal and non-verbal communication was confirmed for the present sample. This supports existing evidence and underlines the importance of working memory in speech diagnostics. Deficits in executive functions should be considered when conceptualising communicative-pragmatic tests for people with dementia-related language disorders. Communication scenarios in the form of role plays are well-suited to investigate a person’s success in everyday communication. Methods to facilitate and enhance communication in these contexts, such as by using personally relevant topics or providing access to real objects, should be considered. As Kindell et al. (2013) showed in their single-case study, even in patients with severe difficulties on neuropsychological assessments, relatively effective communicative strategies may be evident, allowing meaningful conversations [53]. This is especially important if functions such as theory-of-mind or social cognition are impaired. These can occur in both post-stroke aphasia and PPA in the context of dementia and should be considered when examining both disorders.

The data of this study provide no evidence for the hypothesis that especially patients with comorbid depression show negative affect. In fact, P3, who has a comorbid depression, showed fewer affects. This can be explained by a reduced vibratory capacity and flattening of affect. However, the participant showed the most frequent metalinguistic comments. A qualitative evaluation revealed that some of the comments indicate increased frustration and strong distress. These observations show a weakness of the video-based qualitative content analyses in terms of delineating codes and formulating hypotheses. The analysis can only capture visible behavior and verbal utterances, but not the underlying emotions and thoughts of the participants. A questionnaire was used in the superior data collection to assess the satisfaction of the participants. In addition, qualitative interviews after the test could provide information about the participants’ feelings and thoughts during the testing. However, with this vulnerable patient group, the total duration of the appointments must always be kept as short as possible.

The data of this study suggest that the presence of auditory perception disorders is associated with lower linguistic performance. Especially in P1, who has presbycusis, a connection between linguistic and perceptual deficits can be observed. However, it must be considered that, in individual cases, it was not possible to differentiate between a phonological input disorder and an acoustic problem. It was assumed beforehand that perceptual deficits generally influence the selection of modalities. This hypothesis cannot be proven based on the present results, as it requires a larger sample.

During the examinations with the ST, there were no indications of visual disturbances in the participants, neither in the telediagnostics nor in the F2F setting. This shows that the selected cases do not adequately represent the entire population of people with PPA. It can be assumed that there might be patients with visual disturbances influencing their interaction in the ST. There was no visible influence of age in the present sample. The youngest participant (P1) was also the most severely affected. It can be assumed that, given a neurodegenerative speech disorder, the time since diagnosis is more important than the age of a patient.

The years of education of the included cases ranged from 11 to 14 years. P3, who has 14 years of education, was indeed the participant with the lowest verbal deficits, but not the one with the highest total ST scores. Additionally, in relation to the other participants, no major influence can be assumed here, which may also be due to the homogeneity of the cases in terms of education.

The positive influence of supporting relatives could be observed in P1 only. The calming manner and the use of simplified syntax had a positive effect on the communication situation. These observations are in line with the recommendations in the literature in relation to dealing with people with aphasia [47]. Although these observations represent an individual case, it can be assumed that the wife’s participation in several group offers of the Memory Clinic facilitated the handling of her husband’s speech disorder.

The participants described above suffer from different dementia-related language disorders. The document portraits clearly visualise the differences between the participants. A causal relationship with the respective variant of PPA or dementia diagnosis cannot be established. The obvious predominance of executive deficits in the only participant with underlying fronto-temporal lobar degeneration fits the characteristics of this disorder. For the development of communicative-pragmatic procedures for participants with PPA, this means that particular attention must be paid to executive deficits.

The in-depth analysis of three cases that showed differences in modality use between the F2F and the telediagnostic setting allows a profound analysis of the interaction of the individuals. The fact that the code system was adapted to the material shows that a purely quantitative evaluation and testing of hypotheses would not have been sufficient. The openness of the method made it easier to generate new insights. A strong subject orientation enables an orientation towards everyday life since compensatory strategies are specific to every person. In addition, the compensatory strategies of persons with PPA do not only vary between persons, but also for every single person. In a qualitative study with two persons with PPA, it could be shown that compensations are context-sensitive and flexible in their patterns of occurrence [54]. Moreover, the group of persons with dementia-related speech disorders is characterized by a high degree of heterogeneity. In addition to the heterogenic nature of PPA as described at the beginning of the article, most patients are multimorbid, as multimorbidity increases with age [55].

A limitation of the present study lies in the method of video-based transcription and qualitative analysis. Thus, despite careful observation, it was not always possible to determine the cause of a symptom from the observations, as in the case of P1, where it was often not possible to distinguish between executive and verbal deficits. The perseverations, echolalia, and indications of a speech comprehension deficit could also be explained by a disorder of inhibition. This is even consistent with Oksenberg et al.’s (1991) [56] statements on behavior coding.

The work presented here stresses the importance of individual and comprehensive assessments of patients. At the same time, it was shown that there are overarching decisive factors that influence the interaction within the ST. A decision for or against the use of telediagnostics should be made on a case-by-case basis. Orientation for this is provided by the factors identified: extent of executive deficits, extent of language impairment, extent of negative affect, and expression of excessive demands.

## 5. Conclusions

The study points to the potential of qualitative research to explore aspects of telepractice. Our results indicate that especially executive functions, affects, and perceptual deficits need to be taken into account if telepractice is to be applied. Age, education, and distinct forms of dementia might be decisive factors for successful telepractice as well. The extent of effects need to be evaluated in further research. The aid of relatives was not sufficiently illuminated, as only in one of the selected cases was a relative involved in the telediagnostics. In the future, the extent to which deficits in theory-of-mind and social cognition can be circumvented with the help of the latest technology (e.g., virtual reality) for communicative-pragmatic procedures must be investigated. Although the use of virtual reality for people with aphasia is in the exploratory stages of research, a recent review highlights the potential for embedding situated language use into aphasia rehabilitation [57]. Findings of such work could also be used to enhance the diagnostics of communicative-pragmatic skills with assessments such as the ST.

## Figures and Tables

**Figure 1 brainsci-13-00204-f001:**
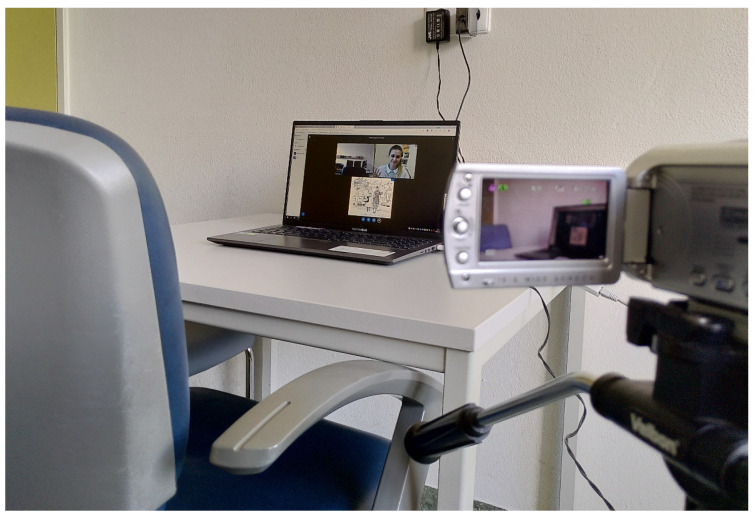
The telediagnostic setting.

**Figure 2 brainsci-13-00204-f002:**
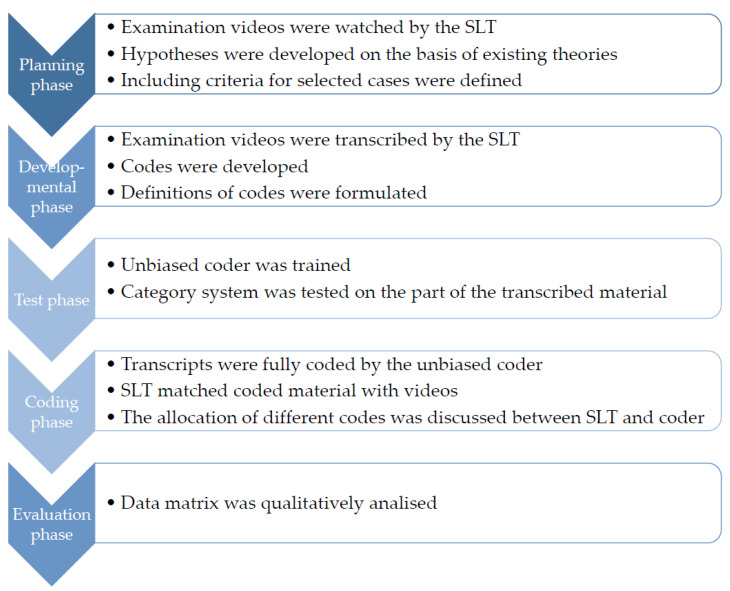
Process of qualitative content analysis according to Kuckartz (2018) [48].

**Figure 3 brainsci-13-00204-f003:**
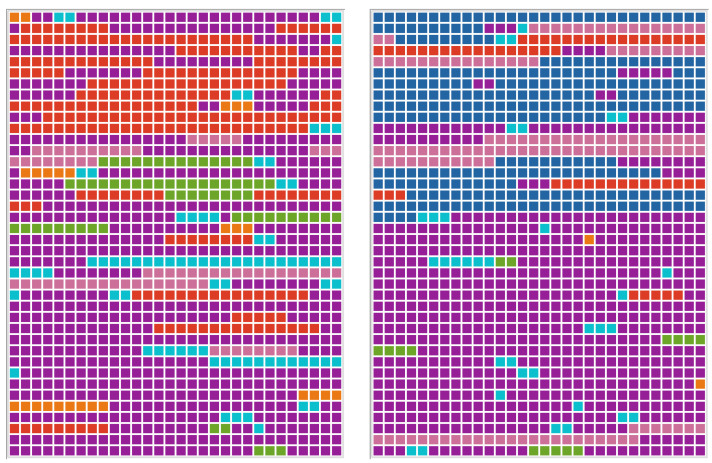
Document portraits of P1–F2F diagnostics (**left**) and telediagnostics (**right**) showing the sequence of codings for the selected document weighted by the length of the segments. Colours of the code system are purple (verbal), pink (expression of excessive demand), red (nonverbal), orange (perceptual), yellow (use of aids), green (executive functions), light blue (affective), and deep blue (aids of relatives).

**Figure 4 brainsci-13-00204-f004:**
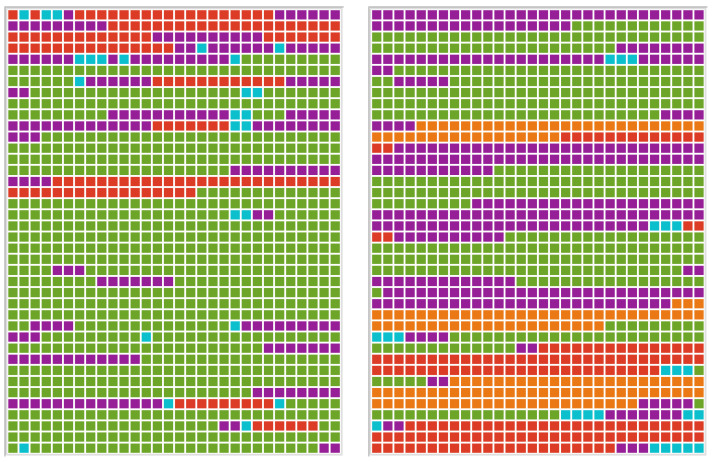
Document portraits of P2–F2F diagnostics (**left**) and telediagnostics (**right**) showing the sequence of codings for the selected document weighted by the length of the segments. Colours of the code system are purple (verbal), pink (expression of excessive demand), red (nonverbal), orange (perceptual), yellow (use of aids), green (executive functions), light blue (affective), and deep blue (aids of relatives).

**Figure 5 brainsci-13-00204-f005:**
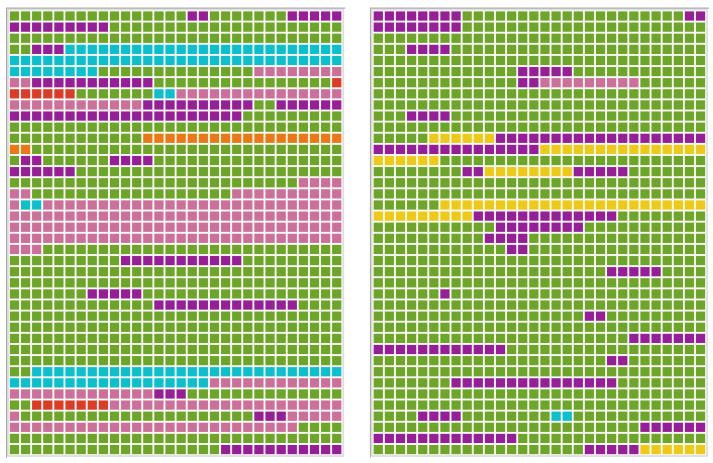
Document portraits of P3–F2F diagnostics (**left**) and telediagnostics (**right**) showing the sequence of codings for the selected document weighted by the length of the segments. Colours of the code system are purple (verbal), pink (expression of excessive demand), red (nonverbal), orange (perceptual), yellow (use of aids), green (executive functions), light blue (affective), and deep blue (aids of relatives).

**Table 1 brainsci-13-00204-t001:** Person-related data of selected cases. nfvPPA = non-fluent variant of primary progressive aphasia; AD = Alzheimer’s disease; svPPA = semantic variant of primary progressive aphasia.

Participant’s Code	P1	P2	P3
Age at start of study	66	79	68
Age at symptom onset	61	76	66
Gender	male	male	male
Years of education	13	11	14
Diagnosis *	nfvPPA	AD	svPPA
Depression *	No	No	Yes
In speech therapy treatment	No	Yes	Yes

* based on medical judgment, including standardized assessments.

**Table 2 brainsci-13-00204-t002:** Category system including codes, definitions, and examples.

Code	Definitions	Examples from the Material
**Verbal**	Aphasic symptoms	“F-Flawless. Seen from that point of view. Yes.” (F2F at P1: 314–314)
**Expression of excessive demand**	Metalinguistic comments that address a heightened awareness of disturbance	“I have everything and then in front of my eyes and then I can’t express it.” (F2F at P3: 190–190)
**Nonverbal**	Use of writing/drawing/gesturing	wiggles hands in the air while exclaiming “eww” (F2F at P2: 152–152)
**Perceptual**	Due to poor sound/image quality; due to presbycusis or presbyopia	“The what, the?” (F2F at P1: 104–104)
**Use of aids**	Compensatory use of objects or media beyond nonverbal communication	makes notes to compensate for deficits in working memory (Telediagnostics at P3: 38–38)
**Executive**	Deficits in inhibition/thinking/planning; Deficits in cognitive flexibility and theory of mind	“Then I would like you to repeat the passage where you were in the sea in the beautiful temperatures in Barcelona. I would like to arrange that. That I can also do [a] tour. (…)” (Telediagnostics at P3: 104–104)
**Affective**	Expressing emotions via meta-linguistic comments/sighing	*sighs, laughs* (Telediagnostics at P1: 60–60)
**Communication partner assistance**	Indications of dependence/ Peculiarities of the communication partners	“That’s okay, honey. Just let it be and listen to what Mrs. Gauch tells you. Don’t do anything about it. Don’t pretend. Okay? Good.” (Telediagnostics at P1: 21–21)

**Table 3 brainsci-13-00204-t003:** Document variables. F2F = face-to-face diagnostics. Tele = telediagnostics.

Participants	P1		P2		P3	
Setting	F2F	Tele	F2F	Tele	F2F	Tele
Verbal	92	97	65	52	23	33
Expression of excessive demand	6	4	0	0	13	1
Nonverbal	19	3	15	6	2	0
Perceptual	5	2	0	4	1	0
Use of aids	0	0	0	0	0	6
Executive functions	6	3	20	12	41	38
Affective	20	18	21	7	4	1
**Communication partner assistance**	0	12	0	0	0	0

## Data Availability

Not applicable.

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
