# Peer review of "Differences of Modality Use between Telepractice and Face-to-Face Administration of the Scenario-Test in Persons with Dementia-Related Speech Disorder"

_brainsci, 2023, doi:10.3390/brainsci13020204_

Round 1

Reviewer 1 Report

Overview of Work:

I commend the authors for their careful analysis of individual task-performance on the ST. This is very important work, especially as it relates to the growing push to examine multimodal functional communication. The detailed narratives provided for each case example, as well as the comparison of the three cases, are particularly strong. As you will see below, my comments address the many questions I had while reading. They are organized by word choice/grammar, content, and citations. Moreover, the overall message of the paper could be strengthened by using clearer and more specific language (I have provided suggestions throughout). Furthermore, there are a few citations missing in the introduction, weakening in the motivation and background knowledge supporting this work. Finally, the methods could use more explicit clarification for future reproducibility, particularly as it relates to the creation and implementation of the coding process. 

Word Choice/Grammar:

1. According to Alzheimer's Disease International (2021) estimated that 55 million people worldwide are affected by dementia and due to demographic change, incidence is 36 expected to increase in future

2. shows an upward trend likely due to increasing awareness of the diagnosticians

3. Symptoms of PPA begin insidiously and develop a severe and progressively global profile

4. PPA can be classified into the non-fluent (nfvPPA), semantic 40 (svPPA) and logopenic (lvPPA) variants

5. The svPPA variant is most frequent, contributing to approximately 39% of all diagnoses 

6.  Murray (1998) suggests that this approach was pleasant for the patient due to its strength-based focus 

7. For patients in late-stage of AD, communicating non-verbally via facial expressions, gestures, or eye contact is recommended

8. Line 103, what is meant by accept? 

9. "It was assumed" is used quite often at the end o the Introduction, but it's not quite sure what is meant by the authors. Are these hypotheses? If so, these statements should be rewritten to reflect this.

10. The final paragraph of the introduction does not end with a clear statement setting up the study to prepare the reader for the Materials and Methods section.

11. Line 145: "inclusion" instead of "including" criteria

12. Figure 1 caption: "an example" should be used here in place of exemplary. Alternatively, just "The telediagnostic setting" suffices.

13. Line 198: "with limited distractions" in place of "furnished with low stimuli"

14. Line 213: using a quantitative analysis procedure

15. Line 235: The use of the word "first" at the beginning of the sentence makes me believe that there is a "second" point coming, which there isn't here.

16. Line 270: "were adapted" in place of "have been adapted"

17. Line 278: "working memory is a constituent of executive functioning"

18. Line 290: "had already been documented" or "observed" instead of "noticed" (same for Line 455 for P3)

19. Line 292: "clinical" instead of "clinic" symptoms

20. Line 293: instead of "proved", we would say "revealed"

21. Line 376: "responsible for employees" instead of "with responsibility" 

22. Line 384: typo, "mnestic" for "amnestic"

23. Line 565: you can end the sentence after "present sample"

24. Lines 564, 578: The word "assumption" is not quite right here. You were testing hypotheses. Similarly, especially given the sample size, the same applies to the term "confirmed". Your data suggests/provides evidence for this outcome, but is not proof of it.

25. Line 587. What is mean by "saturation of data?"

26. Line 636: "might be decisive factors"  (missing the word factors)

27. Lines 638-640: "In the future, the extent to which deficits in theory of mind and social cognition can be circumvented with the help of the latest technology (e.g. virtual reality) for communicative-pragmatic procedures must be investigated."

28. Lines 643-645: "Findings of such work could also be used to enhance the diagnostics of communicative-pragmatic skills with assessments such as the ST"

CONTENT

1. Can the assistance described in Lines 151-153 be clarified?

2. Please provide explicit descriptions of how these codes were defined. Did multiple SLTs work together on this in a group? Were these selected separately by multiple SLTs and then voted upon? I feel that we are missing detail on how these codes were developed and what each code refers to. Examples of each code would therefore also be helpful here! Some of this is touched upon in Figure 2, but Figure 2 needs to be expanded upon in the text.

3. In Line 241, are you referring to a rater blinded to the identity of the communicators or task? Please clarify! Similarly, what is MAXQDA? A citation is likely needed here. 

4. What is an anchor citation? (Line 260)

5. I see now in the Results that you present a very nice Table of the codes referred to in the Methods. I feel that I am missing something: are these the codes previously referred to in the Methods? Or an adapted version? If these are purely the codes developed in advance, it would be very helpful for the reader for this table to appear in the Methods section.

6. In Lines 280-282, you refer to the examiner as the person who decided which codes were assigned to certain features of performance. Was there only one examiner? Is there any inter-rater reliability?

7. You begin a lovely presentation of your analysis for P1 in section "3.3.1. Qualitative analysis in the case of P1", however, you are missing a reference to Figure 3. Without this reference, the color indications in parentheses are confusing. An explanation of the figure content within the main body of the text would also be helpful for the reader.

8. Are these direct quotes from the patients? Are they bilingual? If not, I suggest that you translate them entirely and include the originals in the appendix or not translate them and simply explain the errors in English below each quote. 

9. Line 458: "in the sense of a naming disorder" can be replaced with "consistent with anomia"

10. Lines 462-463: "Thus, multiple criteria for amnestic mild cognitive impairment (MCI) were fulfilled"

11. Line 535: The only participant who needed to or who did seek help from their communication partner? Increasing the specificity here would be helpful!

12. In reflecting upon the codes you adapted to score each participant's performance, it would be more inclusive/generalizable for future work to name #8 "communication partner assistance" as not everyone is going to have a spouse to help them. 

13. Lines 558-561: I entirely agree with your point, but it is awkwardly worded. Here is a suggested edit: "Communication scenarios in the form of role plays are well-suited to investigate a person's success in everyday communication. Methods to facilitate and enhance communication in these contexts, such as by using personally-relevant topics or providing access to real objects,  should be considered." This clarifies the intended message--a message that is very important!

14. What do you mean by the phrase in Line 569? "The participant's affects were not directly influenced." Do you mean to say that the examiner took care to not influence the affect of the participants? If so, this should be explained more clearly and be included in the methods section as well. 

15. Line 626: Lovely point. Can you add in an actual example of a participant behavior that could be coded as either verbal or executive? This would help illustrate your point.

CITATIONS NEEDED

1. Lines 41-47, especially for reported percentages of incidence

2. Line 79

3. Line 81, I suggest adding the following citation: Gallée, J., & Volkmer, A. (2021). A Window Into Functional Communication: Leveraging Naturalistic Speech Samples in Primary Progressive Aphasia. Perspectives of the ASHA Special Interest Groups, 6(4), 704-713. https://doi.org/10.1044/2021_PERSP-21-00021

4. Line 96

5. Line 100, I suggest Dial, H. R., Hinshelwood, H. A., Grasso, S. M., Hubbard, H. I., Gorno-Tempini, M. L., & Henry, M. L. (2019). Investigating the utility of teletherapy in individuals with primary progressive aphasia. Clinical Interventions in Aging14, 453.

6. Line 107

7. Line 245 (MAXQDA)

8. The Discussion section would benefit from citations to other work that has explored conversation analysis in populations with progressive communication disorders. 

Reviewer 2 Report

The authors report three single-cases who were investigated verbal and nonverbal aspects of communication in a face to face and digital setting. Scenario-Test (ST) was performed in person and through video conferencing system. Patients are asked to put themselves in the role of the main person and to communicate as it would be appropriate in the situation. They found that executive functions, affects, and perceptual deficits need to be taken into account if telepractice is to be applied.Age, education, and distinct forms of dementia might be decisive for successful telepractice as well.Although study design is a qualitative study, by coding and transcription into Category system Qualitative data extraction and analysis was also performed. Document portraits of the three patients are clear at a glance. However, due to the small number of cases, when arguing for its representativeness, it may be possible to make limited inferences.

Author Response

Thank you for the appreciative feedback and the time you invested in the reviewing process.

Round 2

Reviewer 1 Report

Thank you very much for your careful inclusion of the proposed revisions!